# Periodic Surface Defect Detection in Steel Plates Based on Deep Learning

**Yang Liu, Ke Xu \***  **and Jinwu Xu**

Collaborative Innovation Center of Steel Technology, University of Science and Technology Beijing, Beijing 100083, China

**\*** Correspondence: xuke@ustb.edu.cn; Tel.: +86-10-62332159

**Abstract:** It is difficult to detect roll marks on hot-rolled steel plates as they have a low contrast in the images. A periodical defect detection method based on a convolutional neural network (CNN) and long short-term memory (LSTM) is proposed to detect periodic defects, such as roll marks, according to the strong time-sequenced characteristics of such defects. Firstly, the features of the defect image are extracted through a CNN network, and then the extracted feature vectors are inputted into an LSTM network for defect recognition. The experiment shows that the detection rate of this method is 81.9%, which is 10.2% higher than a CNN method. In order to make more accurate use of the previous information, the method is improved with the attention mechanism. The improved method specifies the importance of inputted information at each previous moment, and gives the quantitative weight according to the importance. The experiment shows that the detection rate of the improved method is increased to 86.2%.

**Keywords:** periodic defect; deep learning; CNN; LSTM; attention mechanism

## 1. Introduction

Hot-rolled steel plates are widely used in engineering fields such as ship, bridge, machinery, construction, and automobile manufacturing [1]. In the production process of a plate, due to the rolling process, various types of defects can easily form on the surface of the hot-rolled steel plates, such as warping, edge cracking, edge defects, crusting, oxidation, holes, and roll marks. These defects have a great impact on the appearance and performance of the product, so it is extremely important to detect the surface defects on the plate [2].

At present, domestically, surface defect detection is performed mainly by traditional machine learning methods [3,4] and deep learning methods based on convolutional neural network (CNN) networks [5,6]. This method has a high detection rate for non-periodic defects. For example, Yi Li et al. [7] proposed an end-to-end defect detection system for steel strips [7]. In this system, the deep convolutional neural network (CNN) is adopted, the defect image is directly taken as the input, and the defect category is taken as the output. The experiment shows that the detection rate of this method for non-periodic surface defects is above 96%. Di He et al. [8] proposed a new target detection framework called category priority network (CPN) and multi-group convolutional neural network (MG-CNN). In CPN, the image is first classified by MG-CNN. Then, according to the classification result, the feature map group that may contain the defect is separately inputted into another neural network based on yolo (you only look once: a real-time object detection system) [9] for defect location. The average detection rate of this method for non-periodic surface defects is above 96%.

However, the morphological features of periodic defects, such as roll marks, are not fixed. The traditional CNN classifies by extracting morphological features, and the defects with unfixed morphological features are easily misclassified, so that the detection rate is not high.

As periodic defects, such as roll marks, have strong time-sequenced characteristics, long short-term memory (LSTM) [10–12] was introduced, and a periodic defect detection method based on CNN + LSTM was developed. This method combines the advantages of CNN and LSTM. The feature vectors of defect images are first extracted through the CNN, and the extracted feature vectors are inputted into the LSTM for defect recognition.

In view of the non-accurate use of previous effective information of CNN + LSTM, the attention mechanism [13] with the Encoder-decoder framework [14] was introduced to improve the method. This improved method specifies the importance of input information at each previous moment, and gives the quantitative weight according to the importance.

This article has five sections. Section 1 introduces periodic defects and explains the importance of detecting them. Section 2 introduces the features of periodic defects and amplifies the samples to provide data support for subsequent algorithm research. Section 3 introduces the periodic defect detection algorithm based on CNN + LSTM, and then uses the amplified samples to carry-out model training and testing, and analyzes the results. Section 4 introduces the improved detection algorithm with an attention mechanism for the problems in the experiment of Section 3, and experiments using the improved algorithm to verify its effectiveness. Section 5 summarizes the advantages and disadvantages of the algorithm introduced in this paper, and looks forward to the direction of its improvement in the future.

## 2. Description of Periodic Defects

Roll marks on hot-rolled steel plates are typical periodic defects. The training and testing of deep learning networks requires a sufficient number of samples. However, due to the insufficient number of images that were actually collected, we needed manual data amplification. This section introduces features and sample amplification of the periodic roll marks.

### 2.1. Analysis of Features of Roll Marks

Roll marks are a set of uneven defects with periodicity. They are generally due to roll fatigue, insufficient hardness, or foreign matter on the surface of the rolls during rolling operations. The morphological features of roll marks on the same batch are stable and similar; the morphological features of roll marks on different batches can vary due to the repeated rolling of the steel plates. Roll marks are present on both the upper and lower surfaces of plates, mainly at the operating side and the middle position of the plates. They are bright spots observable by the naked eye, and dark spots in the image captured by a camera.

Figure 1a shows an original image of roll marks. Figure 1b shows the defect area and that the roll marks in the frame are arranged periodically. Figure 1c contains details of a single defect, which shows that the morphological features are similar.

Roll marks are sequentially arranged with periodicity. According to the collected data statistics, the morphological features of roll marks on the same batch are similar, as shown in Figure 2. However, the morphological features of roll marks on different batches are very different, as shown in Figure 3.

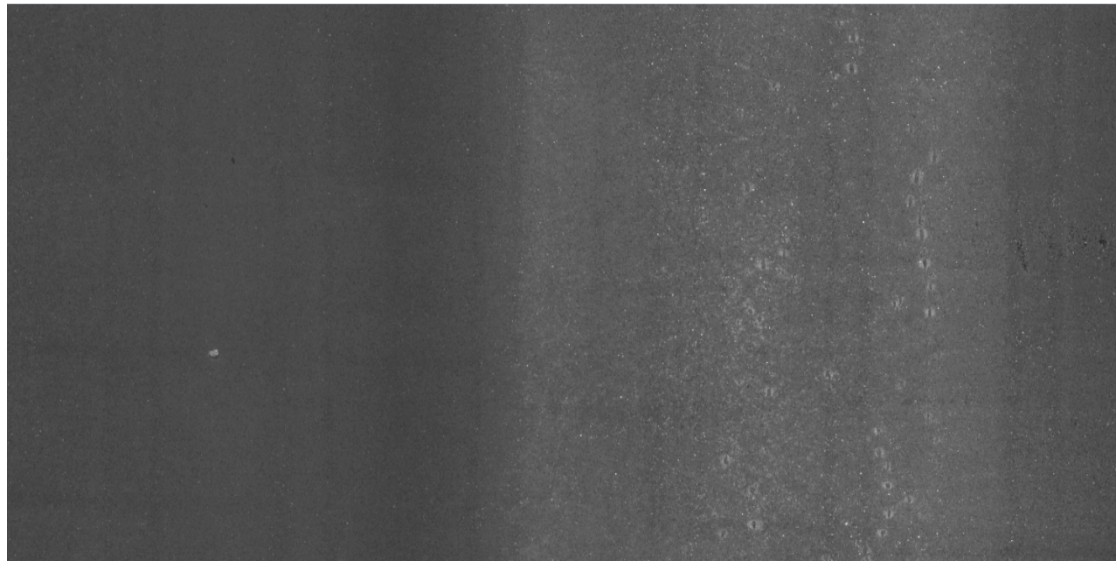

(**a**) Original image

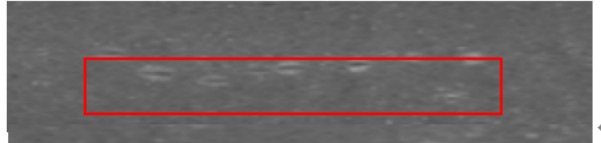

(**b**) Defect area

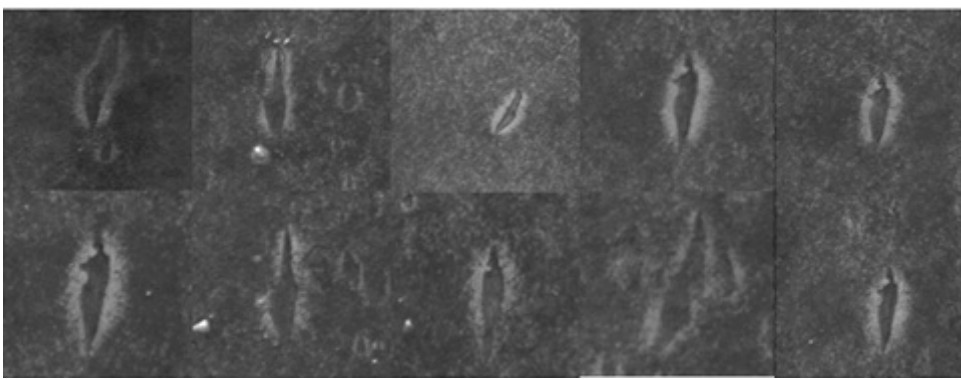

(**c**) Single defect

**Figure 1.** Images of rolling marks.

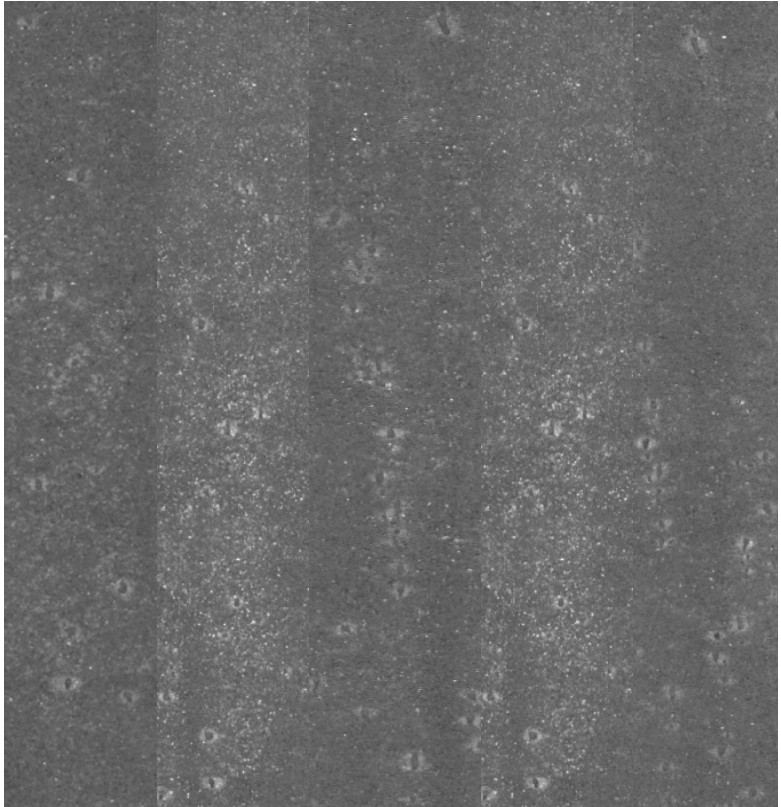

**Figure 2.** Roll marks on the same batch.

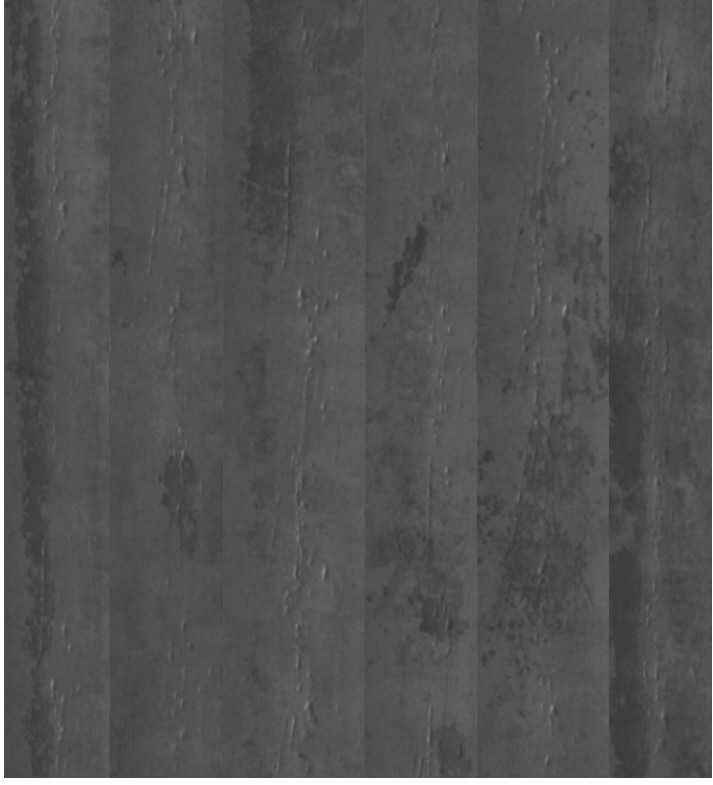

**Figure 3.** Roll marks on different batches.

Roll mark defects are not well-detected because of the greatly different morphological features of roll marks on different batches. The traditional CNN classifies defects by extracted morphological features [6]. Therefore, a CNN can easily misclassify roll marks due to their unfixed morphological features. Consequently, the classification accuracy is not high.

However, as roll mark defects have strong periodicity, their time-sequenced characteristics are suitable for handling by LSTM.

### 2.2. Sample Amplification of Roll Marks

The sample images in the experiment were captured from production lines of hot-rolled steel plates with the surface inspection systems developed by our team. As the number of sample images was limited, for a better model training, a sample generator was designed to augment the number of roll marks to expand the training set.

The generator connected single images of roll marks and backgrounds, and arranged them in a real periodic situation. The generated samples included positive and negative samples.

Positive samples mix the images of roll marks and the backgrounds, and 10 single images were connected to form a long rectangular sample. The occurrence pattern of roll marks on connected single images simulated the occurrence pattern of roll marks on the hot-rolled steel plate, which appears according to a certain periodicity. The generated defect samples were used as a training set. The generated positive samples are shown in Figure 4.

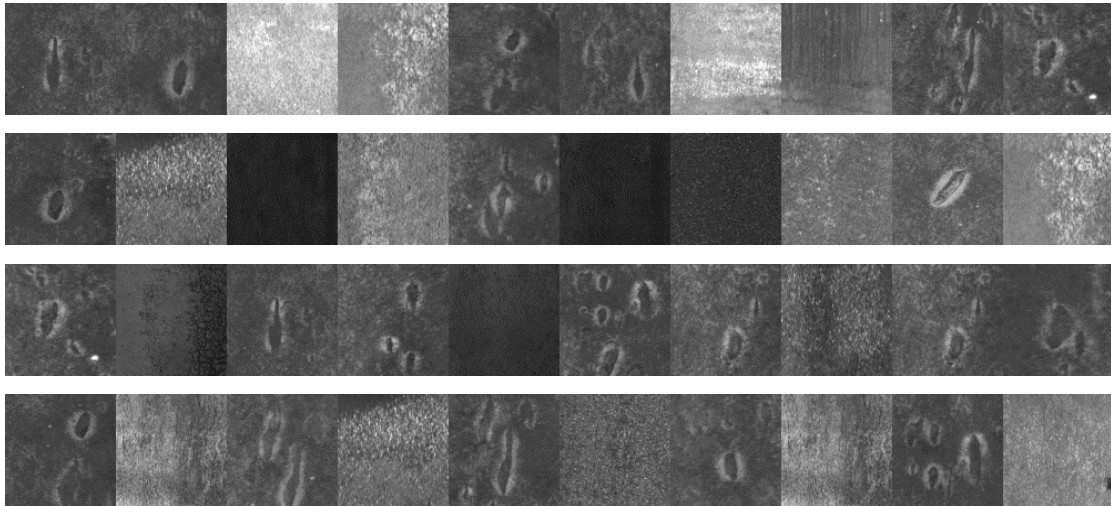

**Figure 4.** Positive samples of simulated periodic defects.

Negative samples only mix the background images, excluding roll marks images. The generated negative samples are shown in Figure 5.

It can be seen that the generated positive samples are arranged with a certain period. If 1 represents a roll mark and 0 represents background, then the periods of positive samples in Figure 4 are: 1100110011, 1000100010, 1011011011, and 101010101, respectively. Defects in different samples have different periods, so the generated samples also have different periods, which make the network learn defects with different periods and enhance the algorithm's robustness.

The generated negative samples are a mixed arrangement of ordinary backgrounds. Because the actual plate images have multiple kinds of interference, we chose background images with various kinds of interference (such as obvious scales) when generating negative samples, which ensured the adaptability of the network to the actual plate images.

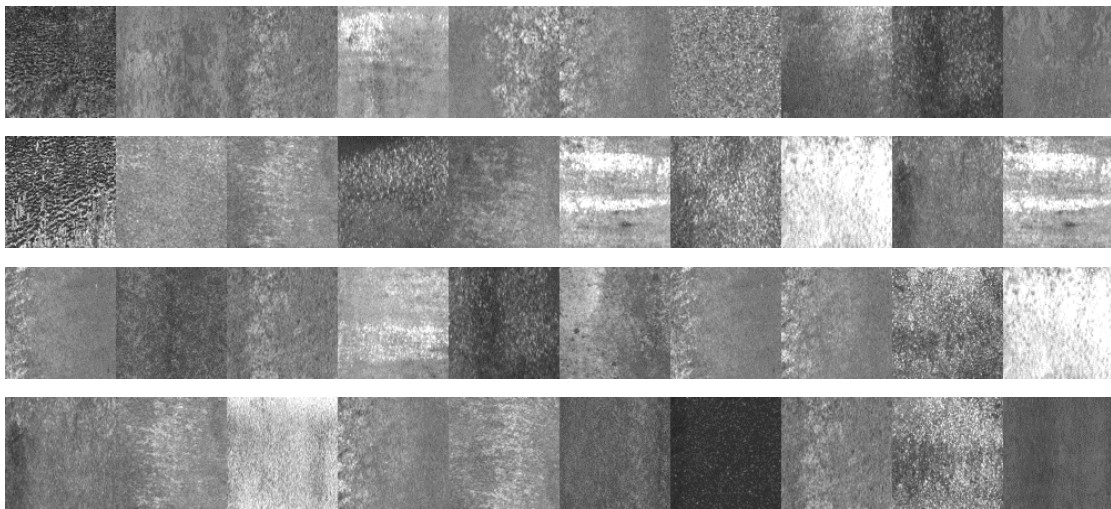

**Figure 5.** Negative samples.

## 3. Periodic Defect Detection Algorithm Based on CNN + LSTM

After generating enough samples, we needed to extract the feature vector with a CNN and use the extracted feature vector as the input of the LSTM to recognize the periodic defects. The following content describes the design and implementation of CNN feature extraction and LSTM defect recognition.

### 3.1. Feature Extraction of a Periodic Defect Based on CNN

The strength of the CNN is that its network structure can automatically learn the features of the input data, and different levels of the network can learn different levels of features, thereby improving the object recognition rate. The CNN consists of three main parts: a data layer, a feature learning network, and a classification network.

The data layer is the input data of the network, and may be an original image or a preprocessed image. The inputted data are sent to the learning features in the feature learning network, and the learned features are classified by the classification network. The CNN in this paper is used to prepare the input feature vectors for LSTM, so the classification network was removed and the output vector of the last maxpooling layer was selected as the input vector of LSTM.

The CNN in this algorithm adopts the VGG16 [15] network and its network structure diagram is shown in Figure 6. Since the network is designed to process the inputted images with a fixed size, all inputted images are sized (224, 224). Since the inputted image is a long rectangular image composed of 10 small images, the size of the inputted image is (224, 2240, 1), and the size of the output feature vector is (7, 70, 512).

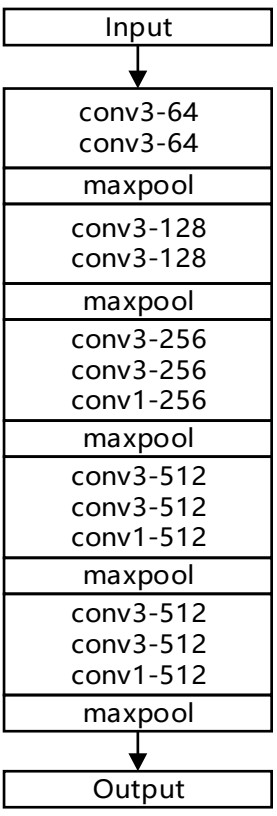

**Figure 6.** Structure of the VGG16 feature extraction network.

*3.2. Periodic Defect Recognition Based on LSTM*

3.2.1. The Principle of Periodic Defect Recognition Based on LSTM

Figure 7a shows the overall flow of CNN + LSTM. The features were extracted from the samples through the CNN to obtain their corresponding feature vectors X. Then, the feature vectors were inputted into the LSTM in a time sequence, and the outputs O are the recognition results.

Figure 7b shows the specific flow of the LSTM [16] algorithm at time t in Figure 7a. In Figure 7b, $C_{t-1}$ is the information of the previous time, including all the information inputted into the network at all previous times. Combined with the information $X_t$ inputted at the current time, the information in the previous time is filtered first, and then the inputted information at the current time is filtered. By filtering, information at times without a defect, such as $X_2$, $X_t$ in Figure 7a, was forgotten, while information at times with a defect, such as $X_0$, $X_{t-1}$ in Figure 7a, was remembered. Thereby, LSTM remembers the defect features better and is prevented from forgetting the defect features for too-long periods.

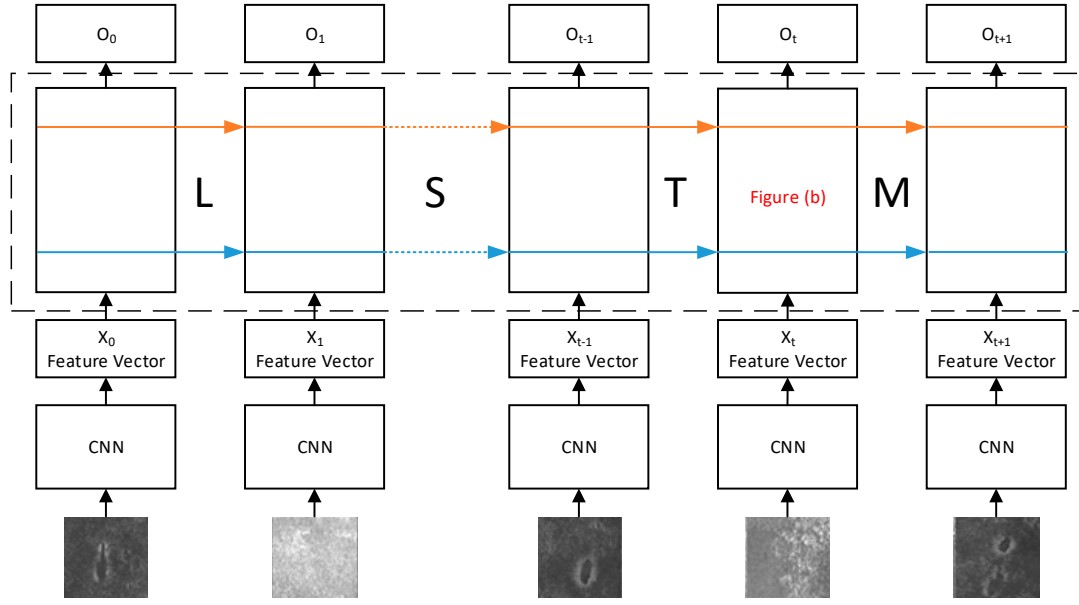

(**a**) **The overall flow of the CNN+LSTM**

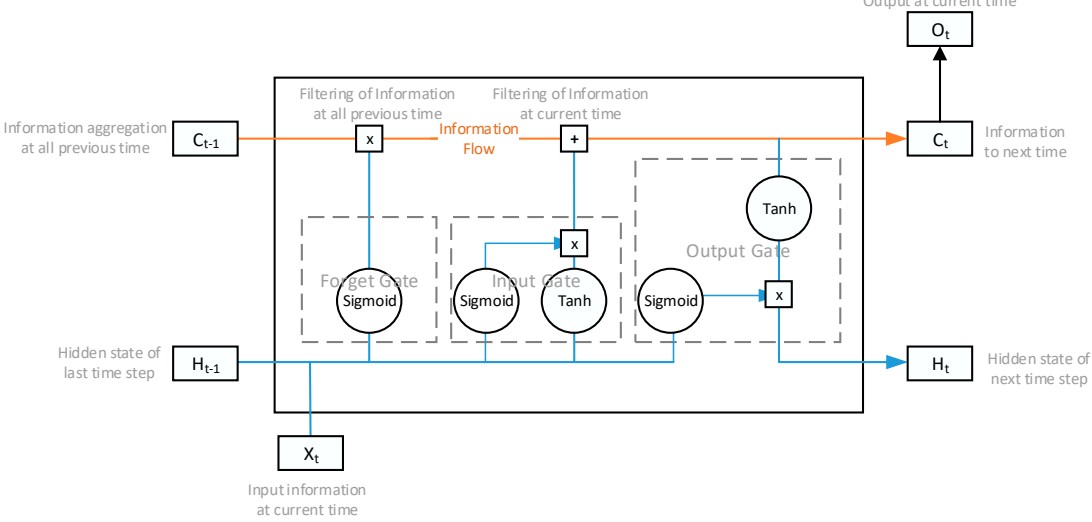

(**b**) **The specific flow of the LSTM algorithm at time t**

**Figure 7.** Flowchart of the convolutional neural network and long short-term memory (CNN + LSTM) detection method.

### 3.2.2. The Specific Process of Recognition of Periodic Defects of LSTM

(1) Constructing an input vector of the LSTM by using the feature vector of the periodic defect extracted by the CNN.

Figure 8 shows a corresponding relation of extracted feature vectors and the original image. A long rectangular image is inputted into the feature extraction network to obtain the feature matrix above. The information contained in one column of the original image corresponds to a multi-dimensional matrix composed of a plurality of columns in the same proportional position. An extracted feature matrix is divided into a plurality of small matrices horizontally, and each of the small feature matrices represents a corresponding matrix in the original image, which is a slicing operation.

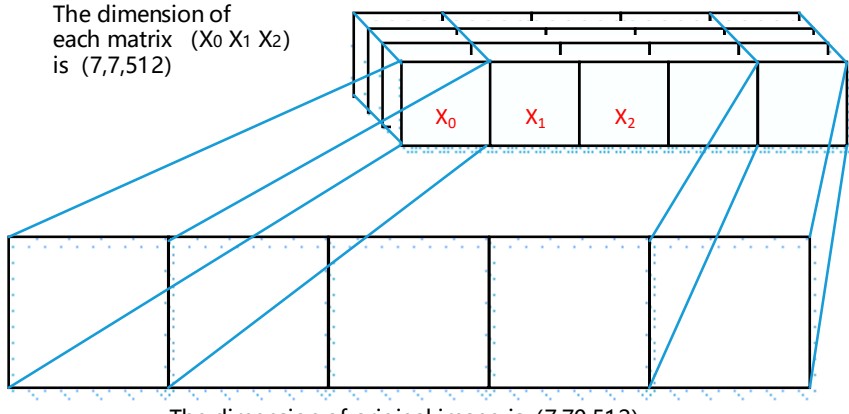

**Figure 8.** Correspondence between extracted features and the original image.

The structure of LSTM is shown in Figure 9. Its input is a set of time-sequenced vectors. The corresponding output is also a set of time-sequenced information. In Section 2.2, the defect image and the background image are periodically arranged and spliced into a long rectangular image to obtain a periodic defect sample. This sample is inputted into the designed CNN, and the output of the convolutional layer is extracted as the feature vector of the sample. Then, the feature vector is sliced into the input vector of LSTM.

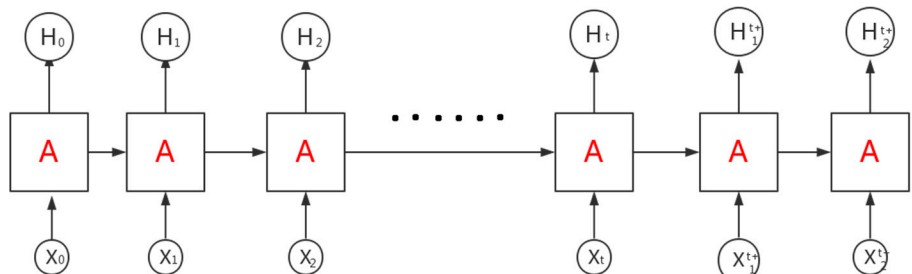

**Figure 9.** LSTM structure.

(2) Training Using LSTM

LSTM was implemented using a Tensorflow [17] framework in this study. The inputted format of LSTM in this framework was {batchsize, timestep, dims}, where batchsize is the training batch; timestep is the total number of time steps (each single defect enters the network at each timestep); and dims is the dimension of the input data.

The dimension of the extracted feature vectors was (7, 70, 512). We sliced it into a set {{7,7*512}, {7,7*512} ......{7,7*512}} of feature vectors as the input of LSTM, i.e., $X_0$, $X_2$ ... ... $X_t$. There are 70/7 = 10 timesteps of features. We added $X_{t+1}$, $X_{t+2}$, timestep = 12, and dims = {7,7*512}.

As shown in Figure 9, the sliced feature vectors were inputted into the LSTM according to the timestep, and the output H of the last cell was the final result. H is a Boolean value. H = 1 means that the input sample is a periodic defect sample, H = others means that the input sample is not a periodic defect sample. The defect recognition network was trained according to the above rules.

*3.3. The Improved Algorithm Based on CNN + LSTM + Attention*

3.3.1. The Principle of the Attention Mechanism

In recent years, attention mechanisms have been widely used in various fields of deep learning [18]. The uniquely human visual attention mechanism is a brain signal processing mechanism: human

vision quickly scans the global image and obtains the focused target area, then pays more attention to this area to obtain more detailed useful information and suppresses other useless information.

The attention mechanism was used in the encoder-decoder framework in this study. However, the attention mechanism can be seen as a general idea that does not depend on an encoder-decoder framework.

The abstract model of the attention mechanism is shown in Figure 10. The elements in Source are a series of <Key, Value> data pairs. An element in the target area is queried. By calculating the similarity between *Query* and *Key$_i$*, each weight coefficient (*Similarity*) corresponding to *Value$_i$* is obtained, and then the weighted sum of *Value$_i$* is calculated to obtain the final *attention* value. So, essentially, the attention mechanism is a weighted sum of the *Value* of the elements in the *Source*, and *Query* and *Key* are used to calculate the weight coefficient of the *Value*. That is, the essential idea can be rewritten as the following formula, where *Lx* = ||*Source*||, which represents the length of the *Source*:

$$Attention(Query, Source) = \sum_{i=1}^{Lx} Similarity(query, Key_i) * Value_i$$

Conceptually, attention can be understood as filtering out a small amount of important information from a large amount of information, and then focusing on the important information and ignoring the unimportant information. The process of focusing is represented in the calculation of the weight coefficient. The larger the weight, the more focus is placed on its corresponding *Value*, that is, the weight represents the importance of the information, and the *Value* is its corresponding information.

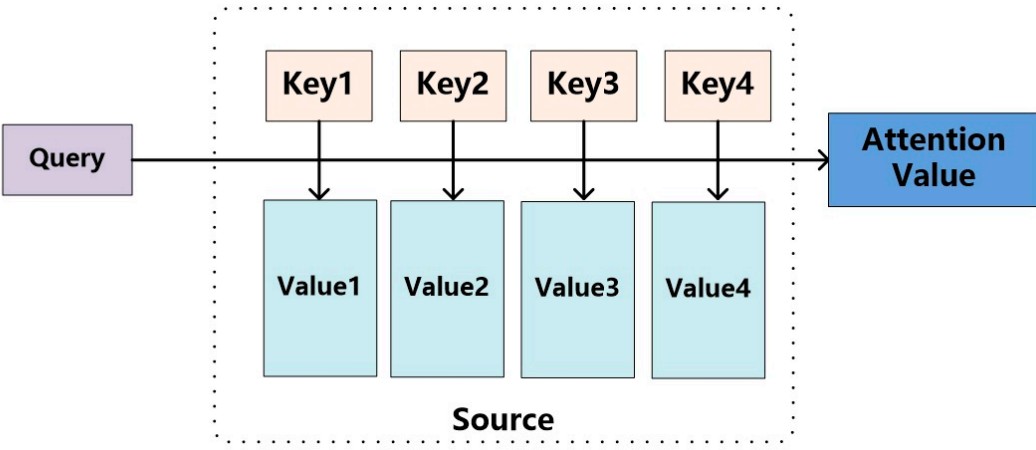

**Figure 10.** Abstract model of the attention mechanism.

3.3.2. The Principle of an Improved Algorithm with the Attention Mechanism

The attention mechanism was implemented in the encoder-decoder framework throughout the network [19]. The encoder-decoder structure without the attention mechanism usually takes the last state of the encoder as the input to the decoder; the subsequent decoder process has nothing to do with the previous input, and is only related to the last inputted state. The encoder-decoder structure with the attention mechanism has different input states of the decoder at different times.

Figure 11 shows the structure of the implementation of the attention mechanism. The specific steps are as follows:

(1) The similarity $a_0^1$ between h$^1$ (the hidden layer output vector of LSTM at the current time) and Z$^0$ (the initial vector, the hidden state of the decoder) is calculated by the matching module "match".

(2) The current output Z$^0$ needs to be matched with each input (h$^1$~h$^4$) to obtain the similarity between the current output Z$^0$ and all inputs ($a_0^1$~$a_0^4$).

(3) Normalize all similarities with the softmax function so that the sum of all similarities (i.e., weights) is 1.

(4)　Calculate the weighted sum of each input ($h^1$~$h^4$) and its normalized weight ($a_0^1$~$a_0^4$), as the next input $c^0$.

(5)　Using $c^0$ as the input of LSTM of the next timestep, the output hidden state $z^1$ of the next timestep is determined by $c^0$ and $z^0$.

(6)　Then, replace $z^0$ with $z^1$ and repeat steps (1)~(5) until the timesteps end.

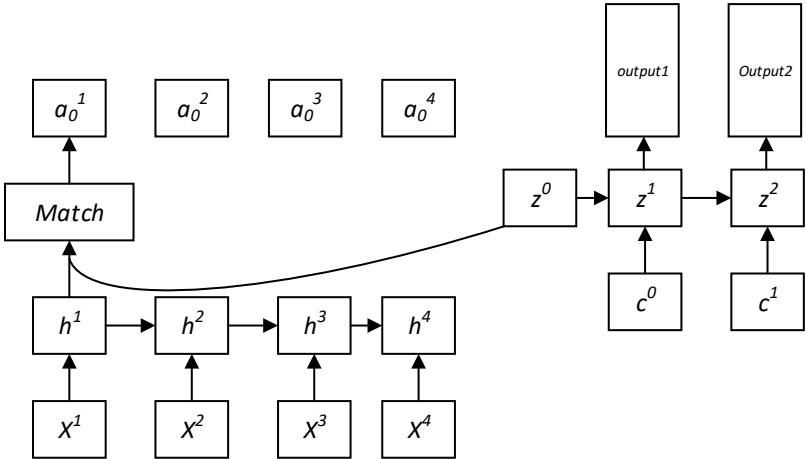

**Figure 11.** The structure of the implementation of the attention mechanism.

By introducing the above attention mechanism into the CNN + LSTM network presented in Section 3, an improved algorithm was obtained, and its structure is shown in Figure 12. First, the CNN extracts the feature of the periodic defect image. Then, the extracted feature vector is inputted into the LSTM with the attention mechanism. Finally, a value is outputted in the last output position. O is the final output, which represents whether there is a periodic defect in this image sequence or not.

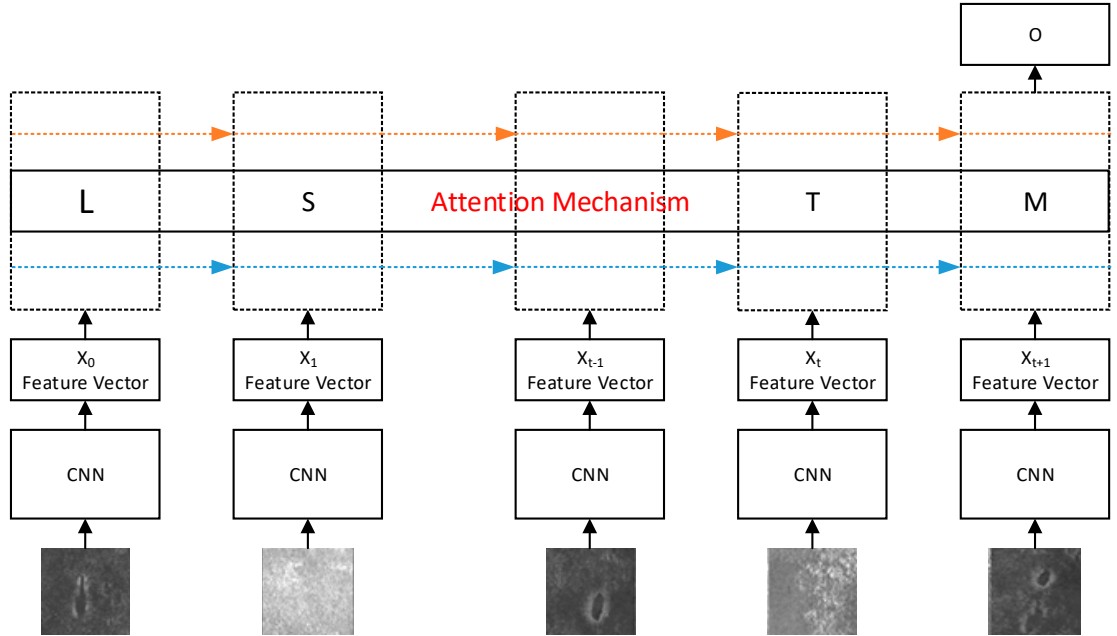

**Figure 12.** The network structure of the improved algorithm with the attention mechanism.

## 4. Experiments

This study used the pre-training model of the VGG16 network provided by Keras without the final fully connected layer. This model, which has tuned network weights, is able to reduce the training time and improve the classification accuracy. We fine-tuned the weights based on this model.

Original real samples (1615 pieces) or amplified samples (15,000 pieces) were used as training sets. During the training, we set the learning rate = $1 \times 10^{-6}$ and the epoch = 20. Twenty percent (20%) of the real samples (323 pieces) were used as the test set. The accuracy of the test set was used for a comparison, where the accuracy of the test set here is defined as: for a given test set, the ratio of the number of samples that were correctly recognized by the classifier to the total number of samples.

Table 1 shows the recognition accuracy of roll marks with VGG16, VGG16 + LSTM, and VGG16 + LSTM + Attention.

**Table 1.** Comparison of VGG16, VGG16 + LSTM, and VGG16 + LSTM + Attention.

|  | **VGG16** | **VGG16 + LSTM** | **VGG16 + LSTM + Attention** |
|---|---|---|---|
| Recognition accuracy with amplified samples | 71.1% | 81.9% | 86.2% |
| Recognition accuracy with original samples | 62.4% | 73.5% | 78.1% |

Table 1 shows that sample amplification obviously improves the recognition accuracy of all algorithms. However, even without sample amplification, both the VGG16 + LSTM algorithm and the improved VGG16 + LSTM + Attention algorithm significantly improved the recognition accuracy.

The recognition rate of the VGG16 + LSTM algorithm is higher than that of the VGG16 algorithm. VGG16 does not perform well because its inputted image is required to be inputted in the size of (244, 244), and some information is lost after the image is compressed. The information on periodic defects is more easily lost during compression due to its low number of appearances, so it is not suitable for periodic defect detection. The VGG16 + LSTM algorithm performs better because LSTM can remember the periodic defect information of the previous time and forget the information without a periodic defect of the current time, which recognizes periodic defects more effectively.

The recognition rate of VGG16 + LSTM + Attention is higher than that of VGG16 + LSTM. The reason for this is that the LSTM mixes the important information inputted at all previous times into the hidden layer state (memory module) at the last timestep. In contrast, the attention mechanism makes clearer that information at which time step is important (the image contains a defect), and provides the corresponding weight according to the degree of importance (the level of similarity between the inputted information at the current time and the hidden-layer state at the last time), which allows the network to better focus on the time when the defect image is input than the time when the background image is input, so that prevents the network from forgetting the defect feature because of the passage of a long period of time.

Next, the trained model is used to detect a real defect. The model performs well on roll marks with little scales. Figure 13 shows the original image of well-recognized roll marks. The roll marks appear periodically and have little scales. This algorithm recognizes them well.

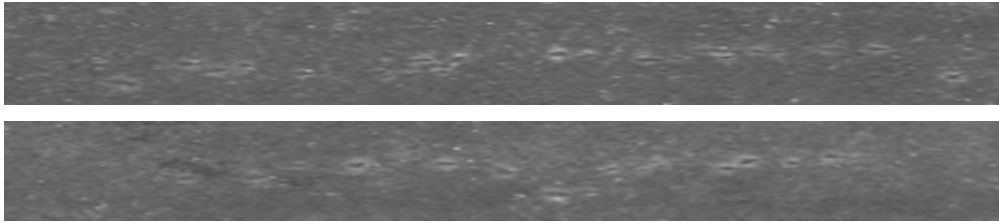

**Figure 13.** Well-recognized roll marks.

When a strong scale appears on the plates, its strong interference results in a low recognition rate. Figure 14 below shows the unrecognized roll marks.

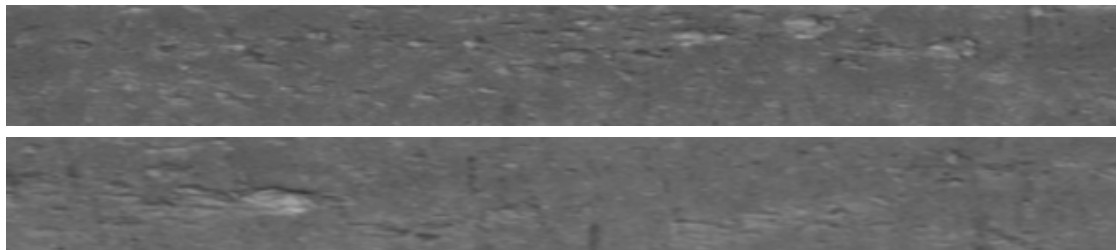

**Figure 14.** Unrecognized roll marks.

## 5. Conclusions

According to the features of periodic roll mark defects of medium and heavy plates, a periodic defect generator was designed and an CNN + LSTM detection algorithm was proposed and tested. Subsequently, an improved detection method with an attention mechanism algorithm was proposed and tested.

The experimental results showed that the periodic defect sample generator simulates real periodic defects well to some extent. The CNN + LSTM + Attention algorithm has a good ability to recognize periodic defects such as roll marks.

Although the algorithm proposed in this study was able to recognize most of the defects, the defect recognition rate with strong scales was found to be insufficient. The attention mechanism also increases the complexity of the algorithm and improves the requirements for computing resources. Our future research will consider using a more powerful feature extraction network and different kinds of training samples.

**Author Contributions:** Conceptualization, Y.L.; methodology, Y.L.; software, Y.L.; validation, Y.L.; investigation, Y.L.; resources, K.X., J.X.; data curation, K.X.; writing—original draft preparation, Y.L.; writing—review and editing, K.X.; visualization, Y.L.; supervision, K.X.; project administration, K.X.; funding acquisition, K.X., J.X.

**Funding:** This research was funded by the National Natural Science Foundation of China, grant number 51674031 and 51874022.

**Conflicts of Interest:** The authors declare no conflict of interest.

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
