# Peer review of "Periodic Surface Defect Detection in Steel Plates Based on Deep Learning"

_applsci, doi:10.3390/app9153127_

Round 1
Reviewer 1 Report
I think the work is interesting and it could be useful. The approach is also good.
I just would like to ask what starting quantities do authors use to extract features? Do authors apply a feature reduction?
Authors didn't made a cross-validation because the had a reference damage?
Author Response
Point 1: I just would like to ask what starting quantities do authors use to extract features?
Response 1:
The starting quantities are 224*2240*1 long rectangular original images. Please refer to the line 140: “the size of input image is (224, 2240, 1),” and input images in bottom of Figure 7(a).
Point 2: Do authors apply a feature reduction?
Response 2:
VGG16 feature extraction network has 5 maxpool layer (a kind of feature reduction) in its structure. Please refer to Figure 6. “Structure of VGG16 feature extraction network.”
Besides, no feature reduction is applied during input of original images into CNN or input of extracted feature vectors into LSTM
Point 3: Authors didn't make a cross-validation because they had a reference damage?
Response 3:
When the data set is small, cross-validation can make full use of limited data to find appropriate model parameters and prevent over-fitting.
If we didn’t make a cross-validation, we only train once but with a cross-validation we train N times and cost N times of training-time.
So when the data set is big enough, cross-validation is not used very often. In this paper, by sample amplification, a sufficiently large data set is obtained.
Reviewer 2 Report
This manuscript proposes an algorithm for detecting roll marks on steel plates. The proposed algorithm is based on CNN and LSTM to extract deep features and use the properties of periodic defects. Furthermore, attention mechanism was employed to improve the detection accuracy. Overall, the contents that use CNN+LSTM and attention mechanism to detect periodic defects are interesting, and it is expected to attract attention of audiences in industrial fields.
However, the quality of presentation is really poor for the publication in Applied Science. At first, the contents for background knowledge (Section 1 and Section 4.1), methods (Section 3 and Section 4.2) and experimental results (Section 3.3 and Section 4.3) are distributed. One-section explanation for the methods will be helpful to well-organize the contents, and it seems to be required to generate “Related work” section to describe background knowledge. There are lots of typos such as missing spacing between words in line 205, and irregular notations for references such as duplicated references in line 29 (ref. 7).
In experiments, definition of the accuracy is required such as recall rate, precision and f1-score. Furthermore, summary for the number of training (before and after data augmentation) and test sets is necessary, and experimental results without the data augmentation will be helpful to highlight the effectiveness of the algorithm.
In short, the manuscript is not ready for publication in Applied Science in this present form. Considerable efforts to revise it are required, and if not then it is better to reject.
Author Response
Point 1: At first, the contents for background knowledge (Section 1 and Section 4.1), methods (Section 3 and Section 4.2) and experimental results (Section 3.3 and Section 4.3) are distributed.
One-section explanation for the methods will be helpful to well-organize the contents, and it seems to be required to generate “Related work” section to describe background knowledge.
Response 1:
The structure of the article was rearranged totally like table of content below
Table of Content
1. Introduction
2. Description of periodic defects
2.1. Analysis of features of roll marks
2.2. Sample amplification of roll marks
3. Periodic defect detection algorithm based on CNN + LSTM
3.1. Feature extraction of periodic defect based on CNN
3.2. Periodic defect recognition based on LSTM
3.2.1. The principle of periodic defects recognition based on LSTM
3.2.2. The specific process of recognition of periodic defects of LSTM
3.3 The improved algorithm based on CNN+LSTM+Attention
3.3.1 The principle of attention mechanism
3.3.2 The principle of improved algorithm with the attention mechanism
4. Experiments
5. Conclusion
References
In which,
(1) All methods were put in Section 3.
(2) All experiments were put in section 4.
(3) For background knowledge, in this paper, the background knowledge (the principle of the proposed new algorithm) and its specific process are described together. Both 3.1 and 3.2 are described in the same way
For example, 3.2.1 is the principle of LSTM algorithm to identify periodic defects, and 3.2.2 is the specific process of LSTM algorithm to identify defects
In order to have a symmetrical structure and facilitate readers' reading, we put the principle of attention mechanism (background knowledge) into section 3.3.1, and 3.3.2 is the specific process of the improved algorithm with attention mechanism.
I think this presentation of putting principles and specific processes together in a continuous sequence more facilitates readers' understanding. If you strongly feel that the "related work" section is needed, please tell me and I will put 3.1, 3.2.1 and 3.3.1 section together as the "related work" section according to your opinion
Point 2: There are lots of typos such as missing spacing between words in line 205, and irregular notations for references such as duplicated references in line 29 (ref. 7).
Response 2:
The typos have been re-examined and completely revised through all this paper. Some typos cannot be revised because they are proper academic noun like “maxpool” “batchsize”.
For new ref. 7, please refer to the line 29 (ref.7).
The reason for missing spacing in line 205 is that “text-align justify” format for paragraph, there are long words in this line so the space between words is expanded by Word.
Point 3: In experiments, definition of the accuracy is required such as recall rate, precision and f1-score.
Response 3:
Accuracy is not recall rate precision or f1-score, its definition is added in experiment section in line 247~249: “The accuracy of the test set here is defined as: for a given test set, the ratio of the sample number correctly recognized by the classifier to the total sample number.”
Point 4: Furthermore, summary for the number of training (before and after data augmentation) and test sets is necessary, and experimental results without the data augmentation will be helpful to highlight the effectiveness of the algorithm.
Response 4:
The number of training (before and after data augmentation) and test sets is added as below.
number of training set before data augmentation | number of training set after data augmentation | number of test set |
1615 pieces | 15000 pieces | 323 pieces |
Please refer to line 245~247 “Original real samples (1615 pieces) or amplified samples (15000 pieces) were used as training sets respectively. During the training, set learning rate=1e-6 epoch=20. 20% of the real samples (323 pieces) were used as the test set.”
Also the experimental results without the data augmentation is presented in Table 1
Table 1. Comparison of VGG16, VGG16+LSTM and VGG16+LSTM+Attention.
VGG16 | VGG16+LSTM | VGG16+LSTM+Attention | |
Recognition accuracy with amplified samples | 71.1% | 81.9% | 86.2% |
Recognition accuracy with original samples | 62.4% | 73.5% | 78.1% |
And corresponding analysis are added in line 253~255 “From Table 1, Sample amplification obviously improves the recognition accuracy of all algorithms. However, even without sample amplification, both VGG16 + LSTM algorithm and the improved VGG16+LSTM+Attention algorithm significantly improve the recognition accuracy.”
Round 2
Reviewer 2 Report
The quality of the manuscript was considerably improved, and the revised manuscript seems to be ready for the publication.